# A Corner Detection Method for Noisy Checkerboard Images

**DOI:** 10.3390/s25103180

**Published:** 2025-05-18

**Authors:** Hui Liu, Ligen Shan, Jiahao Feng, Shuanghao Wang

**Affiliations:** School of Automation, Xi’an University of Posts & Telecommunications, Xi’an 710121, China; liuhui@xupt.edu.cn (H.L.); jhfeng@stu.xupt.edu.cn (J.F.); wang_mouren03@stu.xupt.edu.cn (S.W.)

**Keywords:** camera calibration, corner detection, noisy images

## Abstract

This article proposes a novel approach for corner detection in noisy checkerboard images, comprising several methodical steps: (1) an initial extraction of corners utilizing the cross features present in the edge image of the checkerboard; (2) the elimination of erroneous corners through an analysis of the periodic consistency among the detected corners; (3) the identification of the outermost corners and the subsequent generation of a rectangular bounding box based on the total number of input checkerboard corners; (4) the reconstruction of missing corners, which may have been obscured by noise, by leveraging the periodic characteristics of the corners. Experimental findings indicate that this methodology is capable of effectively detecting all corners of the checkerboard across varying levels of noise, thereby significantly enhancing the success rate of corner detection in noisy images. This makes the proposed method particularly advantageous for camera calibration in special scenarios where noise or contamination in checkerboard images is unavoidable.

## 1. Introduction

Camera calibration represents a fundamental process within the domains of computer vision and image measurement [1]. Its principal objective is to ascertain the intrinsic and extrinsic parameters of the camera, thereby facilitating precise image measurement and three-dimensional reconstruction [2,3]. In the context of camera calibration, checkerboard images have emerged as the predominant calibration pattern, attributed to their regular geometric configuration and the ease with which feature points can be identified [4]. In certain specialized contexts, checkerboard images are susceptible to noise contamination. In applications such as bio-inspired robotic obstacle avoidance and navigation, optical compound eye systems are frequently employed due to their advantages, including a wide field of view and high redundancy [5]. However, the traditional camera calibration method proposed by Zhang necessitates that each ommatidium observe a complete checkerboard pattern [6]. This requirement is often impractical in real-world scenarios due to the structural limitations of compound eyes, where many ommatidia may only capture fragments of the checkerboard or regions affected by noise. Area-array CMOS sensors, which are commonly used in UAV aerial photography for their high frame rates and cost efficiency, inherently introduce imaging noise that significantly impairs the accuracy of feature extraction from calibration patterns [7]. Conventional calibration techniques typically assume that sensor noise is negligible or uniformly distributed. In practice, however, the nonlinear interactions among photon shot noise, dark current noise, and readout circuit noise within imaging systems result in unpredictable grayscale distortions and artifacts in calibration images. High Dynamic Range (HDR) imaging enhances dynamic range by merging multiple exposure images to retain details in both highlights and shadows [8]. Nonetheless, sensor noise, which varies nonlinearly with illumination intensity, can introduce artifacts or distort local contrast in the synthesized images. Neuromorphic vision sensors offer advantages such as high dynamic range and low latency for dynamic scene perception. However, vision-based pulsed cameras are susceptible to generating spatiotemporally correlated noise under interference, leading to traditional corner detection algorithms exhibiting pulse accumulation artifacts or spatiotemporal mismatches [9]. These challenges significantly undermine the accuracy of three-dimensional reconstruction and the reliability of calibration in neuromorphic vision systems. Due to the inherent noise characteristics and motion-induced variations in event data, current feature extraction algorithms based on event cameras remain inadequately developed, resulting in suboptimal performance in feature-oriented methodologies [10]. In contemporary applications such as remote sensing, cartography, and industrial precision metrology, high-resolution cameras have become prevalent due to their high pixel-density sensors. However, their limited single-frame coverage often fails to satisfy large-scale observational needs. By utilizing image stitching techniques to amalgamate multiple images captured by a single camera, a comprehensive panoramic view can be synthesized, facilitating detailed topographic analysis and high-precision dimensional measurements [11,12,13,14]. Modern deep learning-based feature extraction networks utilize end-to-end training mechanisms to adaptively capture multi-scale spatial features [15], demonstrating remarkable robustness in matching under conditions of non-uniform illumination or partial occlusion, thereby significantly improving mosaic quality in complex scenarios. However, when processing checkerboard images characterized by strong periodic structures, the repetitive grid patterns can lead to feature space aliasing, resulting in biases in the estimation of registration parameters, while transposed convolution operations during network decoding can introduce high-frequency artifacts. Coupled with the models’ heavy dependence on annotated data, these limitations restrict algorithmic performance when addressing multi-scale checkerboard patterns. To mitigate these challenges, the intentional injection of structured noise into checkerboard patterns, as depicted in Figure 1, effectively disrupts structural periodicity and enhances spatial feature diversity, thereby creating essential conditions for robust image stitching.

In the context of checkerboard images characterized by substantial noise, the visibility of corner features may be diminished, resulting in potential misidentification or omission of corner locations. This phenomenon can negatively impact the precision of calibration outcomes. Consequently, it is imperative to devise corner detection methodologies tailored specifically for images with high levels of noise.

In recent years, a multitude of algorithms for the detection of checkerboard corners have been developed, including prominent techniques such as Harris and SUSAN [16,17,18,19]. The subpixel-accurate corner detection process utilized in Zhang’s classical calibration method has emerged as the standard in industrial calibration, owing to its high precision and ease of use [20]. This methodology is currently incorporated into the OpenCV library, allowing users to achieve corner detection results and calibration parameters through the straightforward adjustment of relevant thresholds. M.A. et al. [21] proposed a corner detection approach that relies on the accumulation of distances from chords to points, which demonstrates insensitivity to local variations and noise in curves; however, it is prone to detecting false corners in the background. Geiger et al. [22] employed template matching for corner detection, subsequently refining corner positions based on the grayscale gradient, a technique capable of accommodating images with substantial distortion. Yang et al. [23] applied the Hough transform alongside the distribution characteristics of checkerboard lines to identify valid lines, construct checkerboard corners, and initially locate corners. They then utilized a circular template to search for associated points, minimizing discrepancies between these points to achieve precise localization. Wu et al. [24] initially extracted corners based on grayscale distribution characteristics, merging neighboring points at the corners to derive the final coordinates of the checkerboard corners. Nonetheless, this method imposes stringent requirements on image grayscale features, potentially leading to missed detections when corner features are not distinctly visible. Chen et al. [25] employed filters to extract subpixel coordinates of edges, conducted polynomial fitting, and identified edge intersections as corners. Dan et al. [26] utilized the EDLines algorithm for line detection, filtering out background lines through feature filtering, fitting, and optimizing corners based on grayscale gradient sorting, ultimately organizing the corners by transforming the coordinate system. Donné et al. [27] demonstrated that deep learning networks trained on extensive datasets of checkerboard instances achieve detection accuracy comparable to existing algorithms while reducing runtime and maintaining high robustness in the presence of significant noise and lens distortion. Du et al. [28] proposed a method for extracting vertices of adjacent dark squares through binarization and morphological dilation, constructing reference point arrays by calculating geometric midpoints. They subsequently obtained subpixel contour points within the vicinity of reference points using Zernike moments, classified contour points based on row and column orientations and performed bidirectional line fitting to ascertain subpixel corner coordinates, thereby significantly enhancing illumination robustness and localization accuracy. Wang et al. [29] addressed inaccuracies in subpixel-level checkerboard corner detection by introducing a method that integrates local intensity responses with Gaussian–Laplacian responses, effectively improving detection accuracy in noisy environments and under affine transformations.

While the previously mentioned methods for checkerboard corner detection are capable of identifying relevant corner coordinates, they are still prone to both missed and false detections, which complicates the accurate identification of all inner corners of the checkerboard. Consequently, this study introduces a corner detection technique tailored for noisy checkerboard images. The proposed method initially filters the accurate corners by analyzing the local edge features of individual corners alongside the periodic consistency of multiple corner arrays. Following this filtering process, the outermost corners are determined to create a rectangular bounding box. Finally, the method adaptively completes any missing corners by utilizing the inherent periodic characteristics of the checkerboard pattern. This approach effectively mitigates the challenges associated with missed and false detections in corner detection for noisy checkerboard images, thereby significantly enhancing the overall success rate of corner detection.

## 2. Checkerboard Corner Detection

In this section, the analytical process commences with fully stitched images exhibiting substantial noise contamination. The presence of substantial white noise severely compromises the integrity of image features. To mitigate this issue, the present study proposes a systematic corner detection methodology, which is elaborated as follows: (1) Image preprocessing: implement established image processing techniques to diminish the influence of noise on features and to enhance the visibility of the checkerboard pattern. (2) Image corner extraction: identify potential corner locations and compile an initial set of corners. (3) Checkerboard corner filtering: refine the detected corners to retain only those that align with the characteristics of the checkerboard pattern. (4) Standard checkerboard corner grid generation: produce standard checkerboard corners to serve as a reference for subsequent processes. (5) Checkerboard corner merging and completion: address the missing corners by utilizing the generated checkerboard grid and amalgamate closely situated corners to ensure the integrity and precision of the final output. Through the implementation of these steps, the objective is to enhance corner detection efficacy in noisy conditions and to ensure the algorithm’s robustness and adaptability.

### 2.1. Image Preprocessing

In the analysis of the stitched color image, it was observed that following the processes of grayscale conversion and binarization, the corner features were not distinctly visible, and a considerable amount of white noise persisted. Basic filtering techniques, such as Gaussian filtering [30], proved inadequate in effectively mitigating this noise, as illustrated in Figure 2. To enhance the visibility of corner features while minimizing noise interference, morphological operations, specifically erosion and dilation, were employed [31,32]. The erosion operation is particularly effective in eliminating small noise artifacts from the image, particularly those smaller than the defined structural element, thereby improving image clarity; however, it may also lead to a reduction in the size of objects or a loss of finer details. Conversely, the dilation operation serves to recover some of the details that may have been diminished by erosion, while simultaneously augmenting the representation of objects within the image. The application of these two morphological techniques facilitates the enhancement of image clarity alongside noise reduction. In this study, the morphological operations utilized a rectangular kernel with isotropic characteristics, which is particularly advantageous for processing regular checkerboard patterns. Experimental evaluations indicated that larger kernel sizes are more effective in suppressing high-amplitude noise, with a 19 × 19 kernel size being empirically determined to provide optimal results. Following this, the Canny edge detection algorithm was employed to extract the edge features of the image. The Canny edge detection algorithm is noted for its high stability and accuracy across varying lighting conditions, noise levels, and interference, demonstrating considerable robustness [33]. The experimental findings suggest that setting the edge detection thresholds to 0.2 and 0.7 yields more precise and comprehensive edge detection outcomes.

### 2.2. Image Corner Extraction

In the context of corner detection within edge-processed images, the initial identification of corner locations can be based on the characteristics of chessboard corners, which are defined by the intersection of two orthogonal lines [34,35]. To facilitate this identification, a circular sampling approach is utilized. Specifically, a circle with a radius *r* is centered at the origin [36], and sampling points are determined at fixed angular intervals of 0.01 radians across each of the four quadrants to derive their coordinates (x,y). Subsequently, these sampling points from the four quadrants are integrated into a cohesive circular sampling set, as illustrated in Figure 3. The equations for computing the sampling points in the four quadrants are presented as follows:(1)$$\left\{\begin{array}{c} x_{1}=r\sin\theta \\ y_{1}=r\cos\theta \end{array}\right.\;\theta =0,\;0.01,\;0.02,\;\ldots ,\;\frac{\pi }{2}$$(2)$$\left\{\begin{array}{c} x_{2}=r\sin\left(\theta +\frac{\pi }{2}\right)\\ y_{2}=r\cos\left(\theta +\frac{\pi }{2}\right) \end{array}\right.\;\theta =0,\;0.01,\;0.02,\;\ldots ,\;\frac{\pi }{2}$$(3)$$\left\{\begin{array}{c} x_{3}=r\sin\left(\theta +\pi \right)\\ y_{3}=r\cos\left(\theta +\pi \right) \end{array}\right.\;\theta =0,\;0.01,\;0.02,\;\ldots ,\;\frac{\pi }{2}$$(4)$$\left\{\begin{array}{c} x_{4}=r\sin\left(\theta +\frac{3\pi }{2}\right)\\ y_{4}=r\cos\left(\theta +\frac{3\pi }{2}\right) \end{array}\right.\;\theta =0,\;0.01,\;0.02,\;\ldots ,\;\frac{\pi }{2}$$

In the equations, the coordinates of the sampling points are denoted as $\left(x_{1},y_{1}\right)$, $\left(x_{2},y_{2}\right)$, $\left(x_{3},y_{3}\right)$, and $\left(x_{4},y_{4}\right)$, which correspond to the four quadrants. It is essential to configure the sampling radius *r* to be smaller than the pixel width of the individual checkerboard grids present in the acquired image to maintain spatial specificity. However, if *r* is set too small, it may lead to inadequate sampling density, whereas excessively large values of *r* could result in erroneous corner detection due to boundary overreach. This parameter necessitates adaptive modification based on the characteristics of the image, such as grid-scale and noise distribution. Through systematic evaluation, it was empirically established that radii of 20 and 25 pixels provide an optimal balance between sufficient sampling and the minimization of artifacts within our experimental framework.

To analyze the edge-processed image, one must examine each pixel point *P* by aggregating the coordinates of sampling points from the four quadrants, thereby generating a new set of points. This set is then utilized to ascertain the presence of non-zero pixel values within each quadrant. If non-zero pixel values are identified in all four quadrants, pixel point *P* may be provisionally classified as a corner point, with the corresponding radius *r* and angle θ documented. To enhance the precision of the detection process, the characteristic of the chessboard lines is employed, which indicates that points with non-zero pixel values are approximately aligned along a straight line. By varying the radius *r*, the image is traversed once more for evaluation. For the various (r,θ) pairs associated with pixel point *P*, if at least two distinct radii *r* are observed, while the angles θ differ by less than 5 degrees, then pixel point *P* is classified as a corner point, as illustrated in Figure 4a.

In the context of corner detection algorithms, when the physical dimensions of a checkerboard are smaller than the pixel resolution, the edges projected onto the image plane experience sub-pixel shifts due to pixel discretization during rotation. For example, an ideally straight line may be represented as a staircase-like edge that spans multiple pixels, resulting in angular fitting inaccuracies. These inaccuracies become increasingly significant as pixel size increases, particularly when edges occupy a limited number of pixels, leading to angular errors (Δθ) that can reach several degrees. Additionally, when the physical dimensions of the checkerboard corners approach or fall below the pixel scale, edge-averaging effects can produce non-zero pixel intensities across all four quadrants, which may result in erroneous corner detection. To mitigate these issues, the algorithm implements a multi-radius *r* iterative detection approach, coupled with angular consistency verification ($\Delta \theta <5^{\circ }$), to accurately identify genuine corners. By increasing the radius *r*, the algorithm effectively spans a greater number of pixels with edges, thereby diminishing angular sensitivity during individual detection steps. Furthermore, the multi-scale angular consistency check helps to eliminate false corners that may arise from pixel aliasing or noise. This methodology effectively transforms the inherent limitations of pixel resolution into constraints that enhance algorithmic robustness, thereby ensuring the reliable extraction of geometric features in rotated grids, even under conditions of limited resolution.

During the detection process, scenarios akin to those depicted in Figure 4b may arise, wherein precise coordinates for corner points cannot be determined. To mitigate this challenge, the present study introduces a methodology for calculating the centroid, aimed at enhancing localization accuracy and identifying a distinct corner point. A zero matrix H, congruent in size to the edge image, is constructed. For each corner point within the derived corner point set, the values of the 5 × 5 neighborhood matrix surrounding the point in H are designated as 1. These neighborhoods are subsequently labeled and identified as distinct connected components, where the red bounding box in Figure 4c represents one such connected region. The centroid *C* of each connected region is computed, and this centroid *C* is regarded as the accurate corner point. The newly identified corner point $P_{c}$ is documented accordingly. The underlying implementation principle is illustrated in Figure 4c. The formula utilized for calculating the centroid is as follows:(5)$$C=\left(\frac{\sum_{i}x_{i}}{n},\frac{\sum_{i}y_{i}}{n}\right)$$

In this context, $\left(x_{i},y_{i}\right)$ denotes the pixel coordinates within the specified region, while *n* indicates the total number of pixels present in that region. The centroid *C* is regarded as the precise corner point, and the updated corner point $P_{c}$ is subsequently documented. Nevertheless, despite the acquisition of more precise corner coordinates from the edge image, the presence of erroneous corners may persist, as illustrated in Figure 4d. Additional processing is necessary to remove these inaccuracies.

The foundation of the proposed algorithm is its checkerboard-optimized corner detection mechanism, which has been specifically designed to enhance the extraction of checkerboard corners while simultaneously creating an extensive pool of candidates for further evaluation. This design inherently accommodates the presence of two types of artifacts: (1) false positives resulting from texture noise, and (2) non-target corners that display geometric configurations not characteristic of checkerboards. Importantly, although this detection stage exhibits a universal capability for identifying cross-like topological features, it demonstrates limited effectiveness in detecting T-junction or Y-junction corner types due to their unique gradient distribution patterns.

### 2.3. Checkerboard Corner Filtering

In order to identify the chessboard corners from the acquired set of corners, this study utilizes the periodic characteristics inherent to the chessboard configuration. Notably, with the exception of the outermost ring of corners, each internal corner is encircled by four adjacent corners. The vectors that connect a central corner to its four neighboring corners exhibit either perpendicular or parallel orientations. This distinctive feature can be effectively employed to precisely isolate the chessboard corners.

For the designated corner point $P_{c}\left(x_{c},y_{c}\right)$, it is necessary to compute the Euclidean distance $d_{i}$ between $P_{c}$ and each respective corner point $P_{i}\left(x_{i},y_{i}\right)$. Concurrently, one should determine the vector $\vec{P_{i}P_{c}}$ that is established between the current corner point and each of the other corner points. The Euclidean distance $d_{i}$ can be derived using the subsequent formula:(6)$$d_{i}=\sqrt{{\left(x_{i}-x_{c}\right)}^{2}+{\left(y_{i}-y_{c}\right)}^{2}}$$

The vector $\vec{P_{i}P_{c}}$ is calculated as:(7)$$\vec{P_{i}P_{c}}=\left(x_{i}-x_{c},y_{i}-y_{c}\right)$$

Utilizing the acquired set of distances, employ the bubble sort algorithm to arrange the distances $d_{i}$ along with their associated vectors $\vec{P_{i}P_{c}}$. Subsequently, determine the four nearest corner points to $P_{c}$ along with their respective distances, which will be denoted as $d_{1},d_{2},d_{3},d_{4}$, and the corresponding vectors will be represented as $p_{1},p_{2},p_{3},p_{4}$.

Compute the standard deviation σ of the four distances in order to evaluate the proximity of the four corner points, as indicated in Formula (8).(8)$$\sigma =\sqrt{\frac{1}{4}\sum_{i=1}^{4}{\left(d_{i}-\bar{d}\right)}^{2}}$$
where $\bar{d}$ is the mean of the four distances:(9)$$\bar{d}=\frac{1}{4}\sum_{i=1}^{4}d_{i}$$

When the standard deviation σ is less than 10, the four corners are classified as proximal corners, and the current corner $P_{c}$ is documented as a precise inner checkerboard corner. In contrast, if the four corners demonstrate considerable spatial separation, suggesting the lack of four adjacent checkerboard corners surrounding $P_{c}$, then $P_{c}$ is categorized as a non-inner checkerboard corner. The threshold for this classification is established based on the pixel spacing between the checkerboard corners in the image. Given a constant camera focal length, this threshold necessitates only a single initialization for images obtained from a standardized calibration checkerboard, thus ensuring a high degree of adaptability without the need for further modifications.

Designate the vector $p_{1}$ as the reference vector and compute the cosine values $\cos\theta_{i}$ for the angles between $p_{1}$ and each of the vectors $p_{2},p_{3},p_{4}$. Assess whether these vectors are approximately parallel or perpendicular to $p_{1}$. In the event that two vectors are found to be perpendicular to $p_{1}$, labeled as $v_{1}$ and $v_{2}$, alongside one vector that is parallel to $p_{1}$, referred to as $h_{1}$, the corner point $P_{c}$ will be classified as a chessboard corner, as illustrated in Figure 5.

In order to facilitate the subsequent completion of the chessboard corners, the vectors $v_{1}$ and $v_{2}$ are subtracted, and their average is computed to derive the vector $V_{1}$. In a similar manner, the vector $h_{1}$ in conjunction with $p_{1}$ is utilized to obtain the vector $V_{2}$. It is essential to document the vectors $V_{1}$ and $V_{2}$ associated with each chessboard corner, as these vectors will play a critical role in the subsequent processes involved in completing the chessboard corners.

### 2.4. Standard Checkerboard Corner Grid Generation

In the analysis of the chessboard corners, one begins by examining the current corner point denoted as $P_{c}$. If the product of the *x* and *y* components of the associated vector $V_{1}$ is greater than zero, it can be concluded that $V_{1}$ is aligned parallel to the *v*-axis within the pixel coordinate framework. In this scenario, the absolute value of $V_{1}$ is integrated into the vertical vector $B_{v}$, while the *x*-component of the corresponding vector $V_{2}$ is incorporated into the *x*-component of the horizontal vector $B_{u}$. Additionally, the negative value of the *y*-component of $V_{2}$ is added to the *y*-component of $B_{u}$. Conversely, if the product of the *x* and *y* components of vector $V_{1}$ at corner point $P_{c}$ is less than zero, the absolute value of $V_{2}$ is included in the horizontal vector $B_{u}$, and the *x*-component of vector $V_{1}$ is added to the *x*-component of the vertical vector $B_{v}$. Furthermore, the negative of the *y*-component of $V_{1}$ is accumulated into the *y*-component of $B_{v}$, as delineated in Equation (Equation 10).(10)$$\left\{\begin{array}{c} \left\{\begin{array}{c} B_{v}^{\prime }=B_{v}+\left(\right|V_{1x}\left|,\right|V_{1y}\left|\right)\\ B_{u}^{\prime }=B_{u}+\left(\right|V_{2x}\left|,-\right|V_{2y}\left|\right) \end{array}\right.,\;V_{1x}\cdot V_{1y}>0\\ \left\{\begin{array}{c} B_{v}^{\prime }=B_{v}+\left(\right|V_{1x}\left|,-\right|V_{1y}\left|\right)\\ B_{u}^{\prime }=B_{u}+\left(\right|V_{2x}\left|,\right|V_{2y}\left|\right) \end{array}\right.,\;V_{1x}\cdot V_{1y}\leq 0 \end{array}\right.$$

In the equations, $V_{1_{x}}$ and $V_{1_{y}}$ denote the *x* and *y*-components, respectively, of the vector $V_{1}$, whereas $V_{2_{x}}$ and $V_{2_{y}}$ signify the *x* and *y*-components, respectively, of the vector $V_{2}$.

The identification of the four essential corner points of the chessboard is achieved through the comparison of pixel coordinates, which are designated as $P_{u}$ (up), $P_{d}$ (down), $P_{l}$ (left), and $P_{r}$ (right). Subsequently, the mean values of the vectors $B_{u}$ and $B_{v}$ are computed to derive the average vectors $\bar{B_{u}}$ and $\bar{B_{v}}$. These average vectors are then scaled by a factor of 1.5. Utilizing the identified corner points for weighted calculations, the boundary points of the chessboard—specifically $d_{1}$ (top), $d_{2}$ (bottom), $d_{3}$ (left), and $d_{4}$ (right)—are extended and calculated as detailed in Equation (Equation 11).(11)$$\left\{\begin{array}{c} d_{1}=P_{u}-\bar{B_{v}}\\ d_{2}=P_{d}+\bar{B_{v}}\\ d_{3}=P_{l}-\bar{B_{u}}\\ d_{4}=P_{r}+\bar{B_{u}} \end{array}\right.$$

To determine the slope $k_{i}$ and intercept $b_{i}$ for each boundary of the chessboard, one can employ the linear equation representing a line, which is expressed as:(12)$$\left\{\begin{array}{c} \left\{\begin{array}{c} k_{i}=\frac{\bar{B_{v_{y}}}}{\bar{B_{v_{x}}}}\\ b_{i}=d_{i_{y}}-d_{i_{x}}\cdot \frac{\bar{B_{v_{y}}}}{\bar{B_{v_{x}}}} \end{array}\right.,\;i=1,\;2\\ \left\{\begin{array}{c} k_{i}=\frac{\bar{B_{u_{y}}}}{\bar{B_{u_{x}}}}\\ b_{i}=d_{i_{y}}-d_{i_{x}}\cdot \frac{\bar{B_{u_{y}}}}{\bar{B_{u_{x}}}} \end{array}\right.,\;i=3,\;4 \end{array}\right.$$

In the aforementioned equation, $d_{i_{x}}$ and $d_{i_{y}}$ denote the x and y-components of the point $d_{i}$, respectively. Similarly, $\bar{B_{u_{x}}}$ and $\bar{B_{u_{y}}}$ signify the x and y-components of the vector $\bar{B_{u}}$, while $\bar{B_{v_{x}}}$ and $\bar{B_{v_{y}}}$ correspond to the x and y-components of the vector $\bar{B_{v}}$. A central corner point, designated as $P^{*}$, is selected from the established chessboard corners. Subsequently, a weighted operation is conducted on the average vectors $\bar{B_{u}}$ and $\bar{B_{v}}$ to derive the standard chessboard corner $P_{i}^{\prime }$. The formula for this calculation is outlined as follows:(13)$$\begin{array}{cc} P_{i}^{\prime } & =P^{*}+n_{0}\cdot \bar{B_{u}}+n_{1}\cdot \bar{B_{v}},\;n_{0}=-20,-19,\ldots ,20,n_{1}=-20,-19,\ldots ,20 \end{array}$$

The parameter $n_{0}$ must be modified in accordance with the quantity of calibrated checkerboard corners to guarantee that $n_{0}^{2}$ surpasses the overall count of corners. Since the total number of checkerboard corners typically does not exceed 400, this investigation empirically establishes $n_{0}$ within the interval of −20 to 20. As a result, the algorithm is capable of autonomously calculating the necessary corner count without the need for manual intervention, as long as $n_{0}$ adheres to the specified criteria. The value requirement for $n_{1}$ is the same as that for $n_{0}$.

Utilizing the linear equations derived, compute the vertical component *y* for each boundary line in relation to the horizontal component *x* of each corner point $P_{i}^{\prime }$. These vertical components will be designated as $y_{1},y_{2},y_{3},y_{4}$. It is imperative to ensure that the generated network corner points $P_{i}^{\prime }$ remain within the confines of the chessboard by adhering to the specified conditions:(14)$$\left\{\begin{array}{c} \left(P_{i_{y}}^{\prime }-y_{1}\right)\cdot \left(P_{i_{y}}^{\prime }-y_{2}\right)<0\\ \left(P_{i_{y}}^{\prime }-y_{3}\right)\cdot \left(P_{i_{y}}^{\prime }-y_{4}\right)<0 \end{array}\right.$$

In the conditions, $P_{i_{y}}^{\prime }$ denotes the y-component of the corner point $P_{i}^{\prime }$. When these conditions are satisfied, the corner point $P_{i}^{\prime }$ is deemed a valid corner point. By confirming that each corner point $P_{i}^{\prime }$ is situated within the confines established by the linear equations, we can ascertain that the resultant corner points constitute a standard chessboard grid, which accurately reflects the configuration of the chessboard, as illustrated in Figure 6.

### 2.5. Checkerboard Corner Merging and Completion

This research utilizes a grid-constrained corner completion methodology, with the comprehensive decision-making process delineated as follows: initially, a proximity search is conducted for each designated grid point $P_{i}^{\prime }$ within the standard corner grid, wherein the squared Euclidean distance between $P_{i}^{\prime }$ and the identified corners $P_{c}$ from Section 2.2 is computed.(15)$$d_{pc}^{2}={\left(P_{i_{x}}^{\prime }-P_{c_{x}}\right)}^{2}+{\left(P_{i_{y}}^{\prime }-P_{c_{y}}\right)}^{2}$$

In this context, $P_{ix}^{\prime }$ and $P_{iy}^{\prime }$ represent the planar coordinate components of the predetermined grid corners, whereas $P_{cx}$ and $P_{cy}$ denote the coordinate components of the identified corners. The squared distance $d_{pc}^{2}$ is evaluated against a predetermined threshold δ, which is generally assigned a small value; when this distance is less than δ, the two corners are deemed adjacent. In the present study, δ is empirically set to a value of 50. Under these circumstances, the grid corner $P_{i}^{\prime }$ is substituted with the filtered corner $P_{c}$. Conversely, if no adjacent initial corner is identified, the grid corner $P_{i}^{\prime }$ remains unchanged. This approach utilizes geometric substitution and mechanisms for completing missing corners to facilitate corner reconstruction while maintaining subpixel-level localization precision. By systematically iterating through all designated grid points and performing this substitution process, a comprehensive topological structure of the checkerboard corners is ultimately established. The outcomes of this processing are illustrated in Figure 7.

## 3. Experimental Results and Analysis

The apparatus utilized in this study is an industrial CCD video microscope characterized by a resolution of 1920 by 1080 pixels. The optical and lens magnifications are adjustable, with ranges of 21–135× and 0.7–4.5×, respectively. Illumination was provided by an adjustable LED ring light. The camera was equipped with a Japanese Panasonic CMOS image sensor, capable of achieving a maximum image resolution of 48 million pixels and a frame rate of 60 frames per second. High-definition images were transmitted in real-time through a USB interface. This camera is produced by Jiangsu Yunliheng E-commerce Co., Ltd., a company based in Liyang, China, and was acquired through an e-commerce platform. The camera is designed for high-precision measurement applications. Due to the constraints of pixel density in high-resolution cameras, excessively large calibration boards necessitate the capture of multiple images for stitching, which can hinder computational efficiency. Additionally, sparse checkerboard corners may negatively impact the accuracy of subsequent registration and calibration processes. To mitigate these issues, the experiment employs a CG-2 glass checkerboard calibration board with physical dimensions of 15 mm by 15 mm, comprising 17 by 19 corners and grid dimensions of 0.1 mm by 0.1 mm. This configuration, with a pattern accuracy of ±1 μm, adequately meets the dual requirements of spatial resolution and geometric precision necessary for micron-level visual measurement applications. The experimental measurement setup is depicted in Figure 8. The camera was fixed in place, while a 3D micro-positioning stage was utilized to translate the calibration board. Multiple images were captured from a single camera and subsequently stitched using a translation-based stitching strategy. This method preserves the original resolution of individual images, with the final stitched image resolution being contingent upon the resolution of the single images and the dimensions of the overlapping regions. For two images with resolutions of M×N each, if the overlapping region has pixel dimensions m×n, the resulting stitched image resolution can be approximated as (2M−m)×(2N−n). This strategy optimizes the retention of raw image data. Images obtained under varying environmental conditions are stitched, and corner detection experiments are performed based on the resultant stitched images.

### 3.1. Precision Comparison Experiment

To assess the precision of the proposed corner detection methodology, a series of comparative experiments were executed utilizing stitched checkerboard images post-image preprocessing. These experiments benchmarked the new method against Zhang’s calibration technique and three advanced detection algorithms referenced in references [21,22,24]. The truth corner coordinates were established through a manual verification process, wherein human annotators utilized specialized image annotation software to delineate rectangular regions centered on each corner, subsequently extracting the centroid pixel coordinates as reference points. The precision of the detection was quantified by calculating the variance between the coordinates produced by the algorithm and the established ground truth positions across all identified corners, with the mean of these variances serving as the definitive metric for detection deviation.

The comparative outcomes of the five detection methodologies are illustrated in Figure 9. The findings reveal that Zhang’s calibration method demonstrates considerable reliability under standard conditions, achieving a detection accuracy of 98.95% with an almost negligible false detection rate, as detailed in Table 1. However, the method experiences significant performance deterioration in the presence of noise interference; as depicted in Figure 10a, the algorithm encounters a 9.76% missed detection rate and corner coordinate deviations of 0.71 pixels, which severely hampers calibration accuracy. In scenarios involving occlusion, the method exhibits inadequate recognition capabilities for corners obscured by edges, achieving a success rate of less than 40% and lacking effective geometric completion mechanisms. The chord-to-point distance accumulation algorithm referenced in reference [21] is characterized by high sensitivity to noise, resulting in 28.71% false detections in complex backgrounds, as shown in Figure 10b, with detection accuracy diminishing to 71.29% and localization errors escalating to 0.87 pixels. Although the method described in reference [22] maintains a detection accuracy of 97.98%, it suffers from systematic missed detections at the boundaries of the checkerboard, as illustrated in Figure 10c, alongside an increased false detection rate of 2.02% and localization errors of 0.93 pixels. The algorithm presented in reference [24] exhibits marked vulnerability to structural interference from the checkerboard, leading to a proliferation of false corner detections and localization errors that can reach up to 1.30 pixels, as depicted in Figure 10d. In contrast, our proposed algorithm achieves a full detection rate of 98.36% with localization errors of merely 0.67 pixels, representing a 32.1% enhancement over the best-performing baseline method. This outcome substantiates the precision assurance capabilities of our integrated detection and completion framework. Furthermore, under conditions of severe occlusion, the geometry constraint-driven corner completion strategy effectively recovers a majority of the occluded corners while maintaining superior overall detection accuracy compared to existing methodologies, thereby affirming the robustness of the algorithm in occlusion scenarios.

The experiment was conducted using an Intel Core i5-8265U processor, which is equipped with four physical cores and eight logical processors, operating at a base clock frequency of 1.60 GHz. As illustrated in the runtime comparisons presented in Table 1, the algorithms under evaluation demonstrate considerable variations in computational efficiency. Specifically, references [21,22] attain minimal latency through the implementation of lightweight architectures; however, this efficiency is accompanied by significant accuracy-efficiency trade-offs, resulting in precision levels of 71.29% and 97.98%, respectively. The methodology outlined in reference [24], along with our proposed algorithm, exhibit comparable computational durations within the second tier. Nevertheless, our approach, through the application of parallelized optimization, achieves superior detection accuracy while adhering to similar temporal constraints, thereby surpassing the performance of reference [24]. Conversely, Zhang’s calibration method is characterized by the highest computational burden due to its reliance on iterative optimization processes, leading to a latency that is twice that of our method and rendering it inadequate for real-time applications. Importantly, our algorithm sustains sub-pixel localization accuracy while reducing single-frame processing time to below 10 s, thereby achieving an optimal equilibrium between the preservation of precision and the minimization of computational overhead. The proposed architecture successfully meets the real-time operational requirements essential for visual calibration systems.

### 3.2. Lighting Condition Experiment

Variations in illumination significantly influence the luminance and contrast characteristics of images, which in turn directly impacts the efficacy of corner detection algorithms. This study systematically collected and integrated multiple image sets under four distinct illumination conditions: high-intensity lighting, low-intensity lighting, normal lighting, and uneven illumination fields. A thorough evaluation of corner detection was conducted utilizing Zhang’s calibration method, established techniques referenced in references [21,22,24], as well as the proposed algorithm. Figure 11 provides a quantitative comparison of their photometric stability.

The comparative analysis of detection performance among the various algorithms, as presented in Table 2, reveals significant disparities in the robustness of corner detection under complex illumination conditions. Notably, Zhang’s calibration method exhibits stable detection performance under high-intensity, normal, and uneven illumination conditions, effectively identifying checkerboard corners with minimal missed detections and false positives. However, its performance deteriorates markedly in low-light environments, where it struggles to detect valid corners and demonstrates increased sensitivity to lighting variations. The algorithm referenced in reference [21] maintains relatively stable detection performance under normal and strong lighting conditions but experiences considerable performance decline in low-light environments, characterized by a rise in false corner detections. In scenarios of uneven illumination, this algorithm shows inadequate responsiveness in darker regions, resulting in a significant increase in false positive rates. The algorithm from reference [22] achieves the highest detection accuracy under strong lighting and retains full accuracy in low-light conditions, although it experiences minor performance fluctuations under normal and uneven illumination, particularly leading to localized missed detections in dim areas. The algorithm in reference [24] demonstrates optimal detection completeness under strong illumination but suffers from partial, missed detections. While it maintains high accuracy in low-light environments, the number of valid detected corners diminishes. Under uneven illumination, its performance declines significantly, with the correct detection rate dropping to 88.83% and abnormal false positives occurring in regions of abrupt brightness transitions. The proposed algorithm achieves high detection accuracy in uniform illumination scenarios, demonstrating strong adaptability to varying light intensities. In conditions of uneven illumination, the algorithm actively discards candidate points in dark regions that cannot be confirmed as checkerboard areas. Although this strategy results in an 11.76% loss of valid corners, it effectively mitigates false corner judgments and significantly enhances detection reliability. Comparative experiments indicate that the proposed algorithm surpasses existing methods in overall detection accuracy and false positive control, particularly exhibiting greater robustness in complex illumination conditions.

### 3.3. Noise Level Experiment

In order to evaluate the stability of algorithms in practical applications characterized by differing levels of noise interference, we systematically applied controlled noise perturbations to checkerboard surfaces, resulting in the creation of stitched checkerboard images with precisely calibrated noise intensities. A thorough assessment was performed utilizing Zhang’s calibration method, alongside established methodologies referenced in references [21,22,24], as well as the proposed algorithm. The outcomes of the experiments conducted under various noise conditions are presented in Figure 12.

The experimental findings, benchmarked against a ground truth comprising 323 corner points, reveal distinct performance characteristics across different noise levels, with specific detected corner data presented in Table 3. Zhang’s calibration method consistently fails to identify the complete set of corners in the presence of noise, demonstrating a gradual decline in the number of detected corners as noise intensity increases. At noise level III, the algorithm becomes entirely ineffective, resulting in zero valid corner detections due to significant corruption of feature descriptors. Reference [21] exhibits increasing instability with heightened noise levels, maintaining a precision of 74.94% at noise level I, but experiencing a substantial rise in false positives, with precision dropping to 49.84% at level III, suggesting insufficient noise suppression. Reference [22] employs a conservative detection strategy through stringent filtering, achieving perfect precision at level I but failing to detect 50.15% of valid corners. Its precision subsequently decreases to 95.96% at level II and remains at 95.57% at level III, indicating a vulnerability to noise-induced texture confusion. Reference [24] demonstrates moderate robustness, achieving 99.05% precision at level I; however, its detection completeness significantly declines under higher noise conditions. In contrast, the proposed algorithm attains optimal precision at levels I and II through adaptive noise suppression, maintaining a precision of 94.74% at level III while controlling false positives. Importantly, while existing methods exhibit trade-offs between precision and recall or demonstrate sensitivity to noise-induced degradation, our approach maintains a balanced performance through dynamic candidate validation, limiting false positives to less than or equal to 5.26% even in the presence of extreme noise contamination.

### 3.4. Comparative Analysis of Algorithms and Experimental Results

The four corner detection methodologies examined in this study each possess unique advantages and limitations, showcasing specific strengths suited to various application contexts. For instance, Zhang’s calibration technique is characterized by its simplicity and operational ease, making it particularly effective for camera calibration in conventional settings. This method is well-suited for scenarios that demand rapid implementation, cost-effectiveness, and high precision, demonstrating considerable robustness in the presence of low-noise and low-distortion images. The approach detailed in reference [21], which utilizes chord-to-point distance accumulation, innovatively substitutes traditional curvature calculations with distance measurements from edge contour points to chords, thereby diminishing computational complexity. This technique is particularly applicable for corner feature extraction in environments afflicted by severe distortion or substantial noise interference. Reference [22] introduces a robust chessboard corner detection algorithm that leverages template matching and growth strategies. The fundamental premise of this method is multi-stage optimization aimed at achieving high-precision corner localization and reconstruction of chessboard structures. It adaptively identifies chessboards of varying sizes through growth strategies, effectively overcoming the constraints of conventional methods that necessitate predefined chessboard parameters. Additionally, it accommodates scenarios involving multiple chessboards. The methodology presented in reference [24] is tailored for complex environments, detecting chessboard corners by analyzing grayscale distribution characteristics. This approach involves the extraction of candidate corners at chessboard intersections using grayscale features, followed by iterative optimization to eliminate false corners and merge adjacent candidates for precise detection. Our proposed algorithm builds upon the template detection framework of reference [22] by establishing circular sampling points and examining orthogonal corner features. During the verification of orthogonal characteristics, we introduce a predefined angular tolerance range. The subsequent steps adhere to the iterative optimization strategy outlined in reference [24], filtering candidates based on corner arrangement patterns. Furthermore, we integrate angular tolerance into vector angle validation to enhance accuracy in the context of distorted and affine-transformed images. This methodology effectively eliminates false and non-chessboard corners while deriving two orthogonal directional components to create a standardized corner network. Finally, we adapt the corner merging strategy from reference [24] to facilitate the detection and completion of chessboard corners in environments characterized by noise.

Experimental findings indicate that Zhang’s method, along with the techniques presented in references [22,24], demonstrate high detection accuracy across varying levels of noise and illumination. Nonetheless, Zhang’s approach is ineffective in identifying corners under low-light or high-noise conditions, and its computational efficiency is significantly hindered by an excessive number of input corners. The method described in reference [22] encounters challenges with noisy images, leading to considerable feature loss, which results in missed detections and an inability to reconstruct absent corners. Additionally, its performance in template matching deteriorates in dim lighting conditions. Conversely, the technique outlined in reference [24] necessitates substantial grayscale differences at chessboard intersections and primarily concentrates on eliminating false corners, which consequently reduces corner counts in the presence of noise or variations in illumination. The method referenced in [21] is capable of detecting corners under a range of illumination and noise scenarios; however, it tends to produce an excessive number of false corners due to inadequate validation criteria, resulting in insufficient precision for vision measurement applications. In contrast, our proposed algorithm successfully achieves accurate corner detection and completion across diverse illumination and noise conditions while preserving precision. In situations characterized by severe noise or occlusion, where fewer than half of the anticipated corners are detected in a boundary row or column, our algorithm prioritizes measurement accuracy by selectively discarding uncertain corners rather than attempting to reconstruct ambiguous boundaries.

The experimental results suggest that while the algorithm presented in this study demonstrates superior performance in corner detection and completion capabilities compared to existing algorithms, particularly in noisy environments, it exhibits a significant reliance on prior knowledge of the checkerboard structure. This limitation confines its applicability primarily to regular network calibration targets. The accuracy of corner completion is notably compromised when the checkerboard experiences physical deformations, such as bending or folding. Moreover, the algorithm struggles to detect and complete all corners effectively in scenarios characterized by occlusion and uneven lighting. The employed corner discarding strategy fails to account for the continuity and spatial arrangement of local corners, which may result in the inadvertent loss of information from certain detected corners during the delineation of the checkerboard boundary. In instances where the checkerboard structure within the stitched image is incomplete, the presence of a completion strategy does not facilitate the creation of a standard network beyond the image, thereby leading to algorithmic inaccuracies. Additionally, the algorithm does not address the potential coexistence of multiple checkerboards; when multiple checkerboard structures are present within a single image, it may erroneously determine the network boundary and miscalculate the horizontal and vertical components of the corners. This miscalculation can lead to confusion within the corner network, ultimately undermining the reliability of the algorithm’s detection results.

## 4. Conclusions

This study presents a method for checkerboard corner detection and automatic completion specifically designed for noisy stitched images, offering a viable solution for the calibration of high-resolution cameras. By leveraging information from multiple images and intentionally introducing noise during the stitching process to create complete checkerboard patterns, the proposed approach achieves high-precision corner detection and completion in high-noise environments. Experimental findings indicate that the algorithm demonstrates superior detection accuracy, exhibiting reduced corner localization errors and significantly enhanced precision in comparison to existing techniques. Furthermore, tests conducted under varying illumination conditions validate the robustness of the method, effectively addressing common lighting variations encountered in practical applications. Future research will aim to tackle the existing challenges associated with completing edge corners in scenarios characterized by non-uniform illumination and partial occlusion, with the goal of improving overall detection performance and adaptability to specialized conditions.

## 5. Author Statement

The language expression in this manuscript has been optimized with the assistance of editing tools, but all research content, data analysis, and academic conclusions are the original work of the authors.

## Figures and Tables

**Figure 1 sensors-25-03180-f001:**
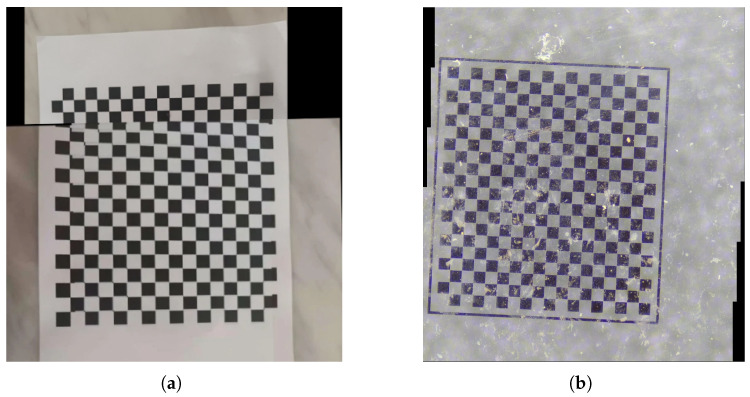
Comparison of stitching results between noise-free and noisy images. (**a**) Noisy-free. (**b**) Noisy.

**Figure 2 sensors-25-03180-f002:**
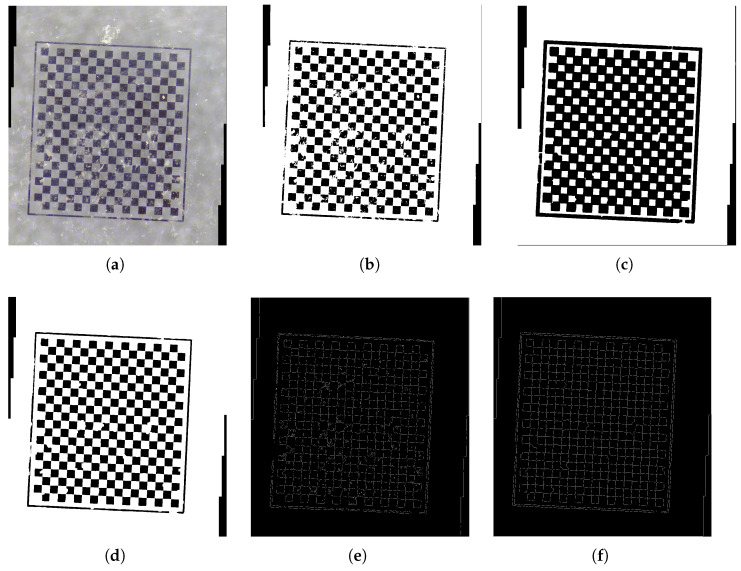
Comparison of the effects of different image processing techniques. (**a**) The origin mosaic image. (**b**) Gaussian filtering. (**c**) Dilation. (**d**) Erosion. (**e**) Edge detection after Gaussian filtering. (**f**) Edge detection after dilation and erosion processing.

**Figure 3 sensors-25-03180-f003:**
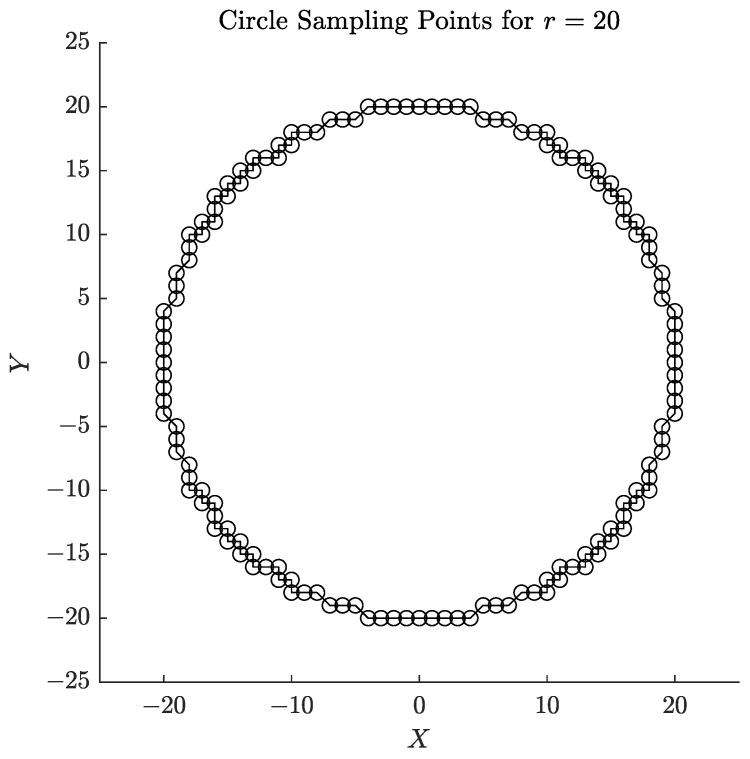
The coordinate system image of circular sampling points.

**Figure 4 sensors-25-03180-f004:**
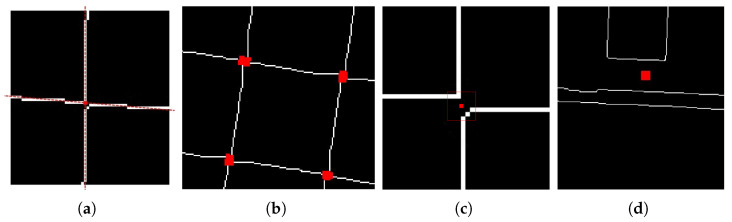
(**a**) Corner detection pipeline. (**b**) Raw points. (**c**) Centroid calculation. (**d**) False positive examples.

**Figure 5 sensors-25-03180-f005:**
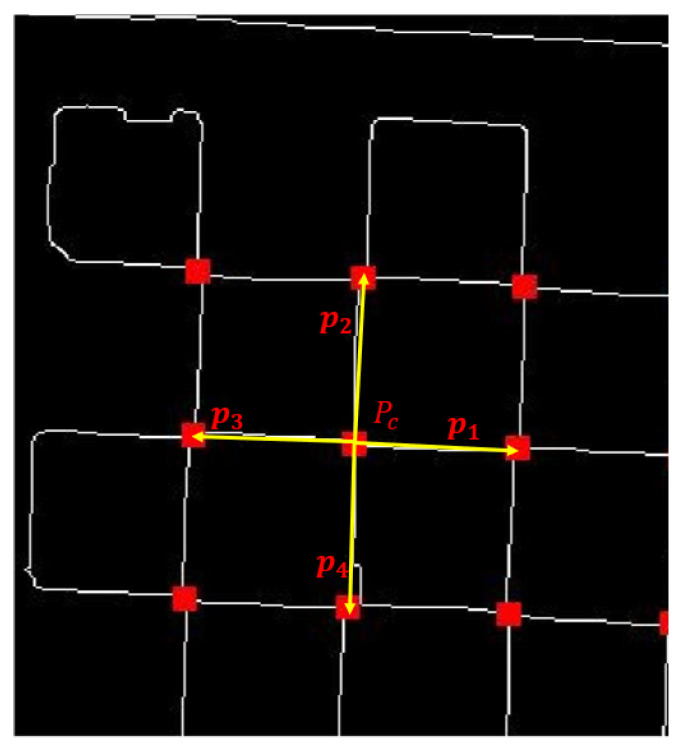
The schematic diagram of the checkerboard corner screening process.

**Figure 6 sensors-25-03180-f006:**
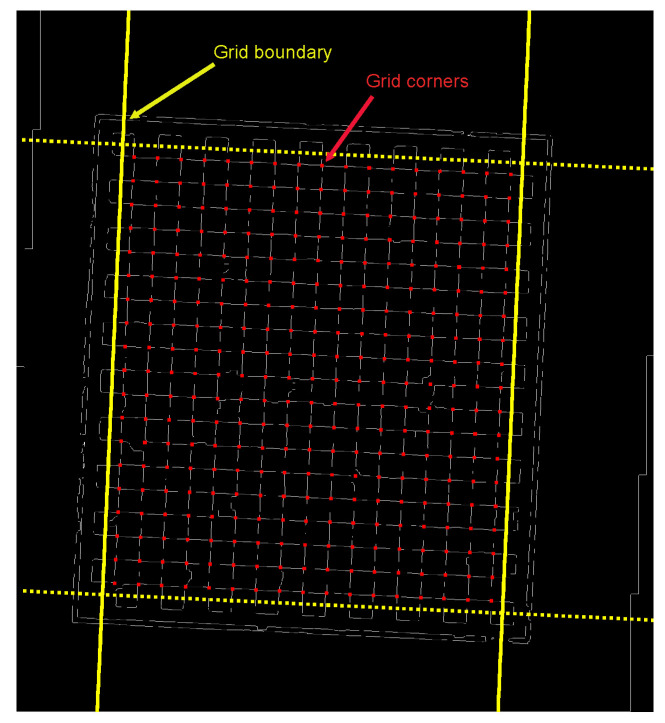
Standard checkerboard corner grid.

**Figure 7 sensors-25-03180-f007:**
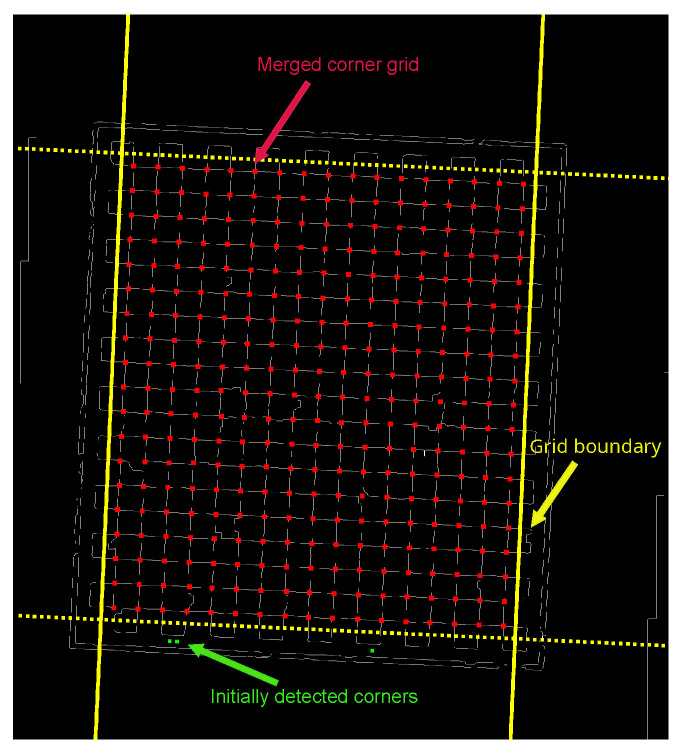
Standard checkerboard corner point network diagram.

**Figure 8 sensors-25-03180-f008:**
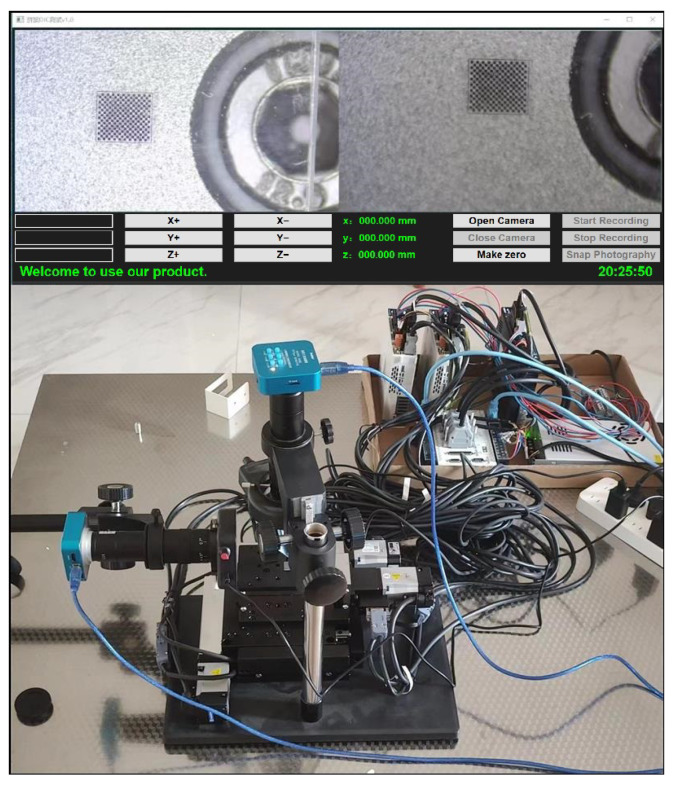
The corner detection experimental setup.

**Figure 9 sensors-25-03180-f009:**
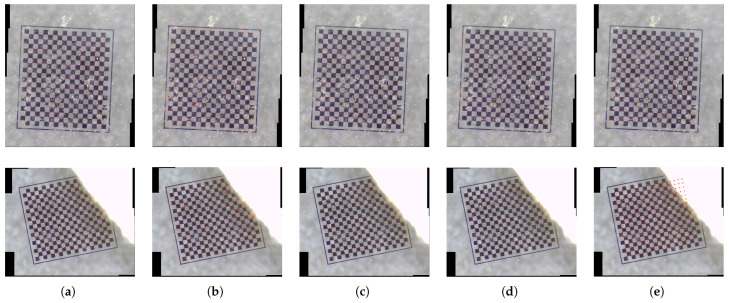
Detection performance comparison: non-occluded cases and occluded conditions. The red dots depicted in the images represent the identified coordinates of the corners. (**a**) Zhang. (**b**) Ref. [21]. (**c**) Ref. [22]. (**d**) Ref. [24]. (**e**) Ours.

**Figure 10 sensors-25-03180-f010:**
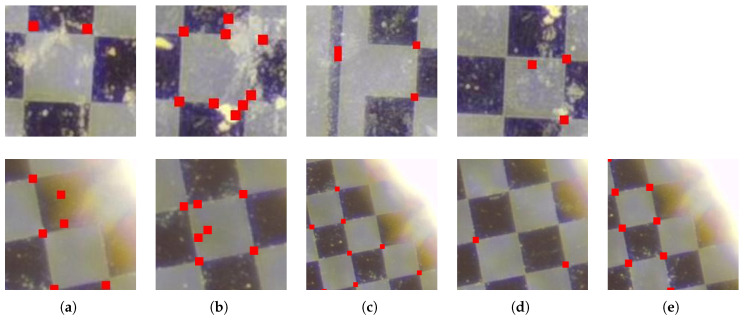
Summary of issues in corner detection under non-occluded and occluded conditions. The red dots depicted in the images represent the identified coordinates of the corners. (**a**) Zhang. (**b**) Ref. [21]. (**c**) Ref. [22]. (**d**) Ref. [24]. (**e**) Ours.

**Figure 11 sensors-25-03180-f011:**
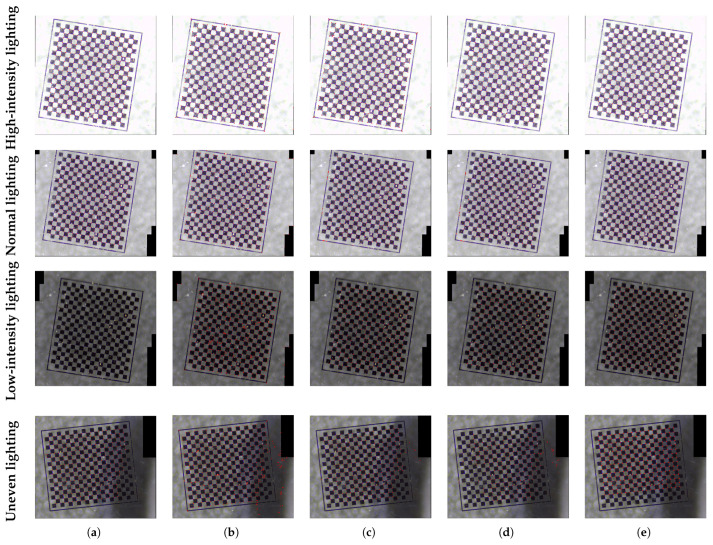
Detection results under different lighting conditions. The red dots depicted in the images represent the identified coordinates of the corners. (**a**) Zhang. (**b**) Ref. [21]. (**c**) Ref. [22]. (**d**) Ref. [24]. (**e**) Ours.

**Figure 12 sensors-25-03180-f012:**
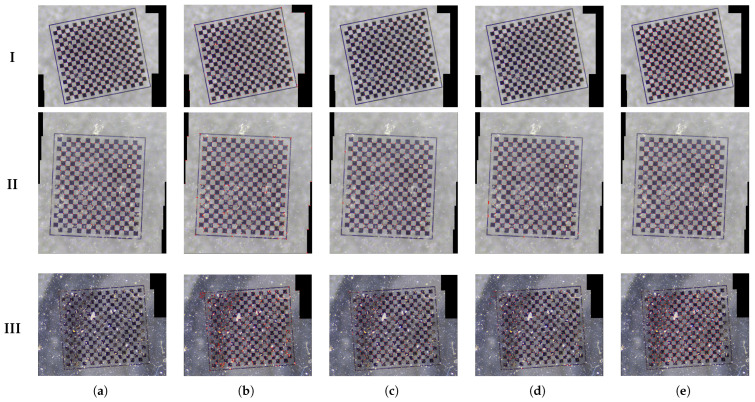
Detection results under different noise levels. The red dots depicted in the images represent the identified coordinates of the corners. (**a**) Zhang. (**b**) Ref. [21]. (**c**) Ref. [22]. (**d**) Ref. [24]. (**e**) Ours.

**Table 1 sensors-25-03180-t001:** Comparative analysis of corner detection methods.

Algorithm	Precision (%)	Recall (%)	Bias (pixel)	Average Runtime (s)
Zhang	98.9547	90.2476	0.71278	16~17
Ref. [21]	71.2992	95.356	0.87669	4~5
Ref. [22]	97.9876	72.6006	0.9348	5~6
Ref. [24]	95.0743	84.5201	1.3075	9~10
Ours	98.366	94.7368	0.66636	8~9

**Table 2 sensors-25-03180-t002:** Summary of corner detection counts under varying illumination conditions.

Algorithm	Correct Corners/Detected Corners			
	High-Intensity Lighting	Normal Lighting	Low-Intensity Lighting	Uneven Lighting
Zhang	323/323	314/317	—	301/303
Ref. [21]	323/419	316/420	322/484	305/470
Ref. [22]	311/312	288/299	257/257	273/277
Ref. [24]	295/297	274/317	225/230	191/215
Ours	323/323	323/323	323/323	285/285

**Table 3 sensors-25-03180-t003:** Summary of corner detection counts under varying noise levels.

Algorithm	Correct Corners/Detected Corners		
	Noise Level I	Noise Level II	Noise Level III
Zhang	314/316	299/299	—
Ref. [21]	323/431	316/467	321/644
Ref. [22]	161/161	310/323	216/226
Ref. [24]	209/211	297/328	205/228
Ours	323/323	323/323	306/323

## Data Availability

Data are contained within the article.

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
