# Peer review of "A Corner Detection Method for Noisy Checkerboard Images"

_sensors, 2025, doi:10.3390/s25103180_

Round 1
Reviewer 1 Report
Comments and Suggestions for Authors
This manuscript proposes a novel corner detection method tailored for noisy checkerboard images, leveraging checkerboard feature constraints to ensure high-precision corner detection across varying noise levels. Overall, this is a good paper. However, the collation of manuscripts is far from meeting the requirements of journal publishing. Before further consideration of the manuscript, several problems need to be solved.
- In the Introduction, the author mentions "In this case, artificially adding noise... " but fails to elaborate on the specific application scenarios. The author is requested to provide a detailed explanation of the applicable scenarios of the proposed method and include more references to support it.
- The corrosion expansion process in the Image Preprocessing section is highly sensitive to the choice of scale. The author should explain how the method in this paper determines the scale of morphological filtering.
- In the Extracting image corner, there is no discussion on closely related parameters such as the sampling radius. Additionally, the method tends to produce numerous false detections. The author needs to analyze the robustness of this method.
- There are numerous formatting errors in the manuscript, such as spaces after periods and images that exceed the page size and lack clarity. The author is asked to place vector maps.
- The main highlight of the article is the proposed anti - noise corner detection algorithm. However, the experimental part does not discuss the algorithm's performance under different noise levels. It is hoped that the author will add a comparative test of noise levels.
-
This paper provides a systematic introduction to relevant methods, and now there are many brain-based vision sensors that need corner detection method. Suggest introducing some work on brain-based sensors like event cameras and spking cameras, such as robust decoding of rich dynamical visual scenes with retinal spikes, and stereo event-based, 6dof pose tracking for uncooperative spacecraft. This will allow the proposed method to have more application scenarios.
- In addition to the algorithm mentioned in the article, it is necessary to compare some classic algorithms, such as the widely recognized Zhang Zhengyou chessboard extraction algorithm, to enhance the persuasiveness of this article.
- For the extraction accuracy comparison experiment mentioned in this paper, the author is asked to explain how to obtain the true value of the diagonal mark position.
- Figure 11 shows the extraction effect of the algorithm under different illumination conditions. The author should clarify whether the algorithm remains stable under uneven illumination.
- This paper mainly uses the extracted corner points and the corner points complemented by the surrounding diagonals to achieve stable positioning of the checkerboard. The author is asked to answer the following questions:
1 ) Is the accuracy of the complemented corner points reliable, and will it reduce the extraction accuracy?
2 ) How does the author choose between corner extraction and corner completion when they coexist?
3 ) Is the completion method also robust to occlusion imaging?
- I think the applicable conditions of this method are very limited, and it is not robust for cases with affine transformation and lens distortion. Additionally, the author does not mention the computational efficiency of the algorithm. The author is asked to further analyze the innovation and robustness of the algorithm.
- Currently, there should be many high - precision and high - robust checkerboard extraction algorithms. The author is invited to conduct a comprehensive investigation and increase and update the references.
Reviewer 2 Report
Comments and Suggestions for Authors
The article presents an effective corner detection method specifically designed for noisy checkerboard images. The corner detection was developed for a relatively narrow task, where a periodic structure with orthogonal angles is given. The authors state, "It is suitable for camera calibration in situations where noise or contamination in checkerboard images cannot be avoided," but if it is necessary to calibrate a system (a camera), one would typically choose optimal conditions, calibration object, lighting, etc. The article is well-written and understandable for the reader. However, given the aforementioned points, I would suggest publishing this article in another, more specialized journal that focuses on solutions to such problems.
I would also like to provide some comments for improving the article:
I would recommend touching upon artificial intelligence methods in the review of detection techniques. The problem described in your article is one of the fundamental tasks addressed with the help of artificial intelligence (the segmentation task).
In my opinion, it is necessary to indicate how the accuracy of angle determination correlates with the pixel sizes of the camera. If the grid is rotated, there will be an error in attributing the line (angle) to one of the pixel lines of the detector. Or, in your case, are the identified objects much smaller than the resolution provided by the camera?
In Table 1, your method shows the highest results for angle determination compared to other methods. It is impossible to evaluate how much better it is than the others; it is necessary to find the conditions under which your method will not yield a 100% accuracy. Perhaps your method achieves a 99% accuracy, while other methods show less than 50%. In your method, all angles are always determined; up to what point (level of illumination) will this continue to happen?
Reviewer 3 Report
Comments and Suggestions for Authors
This manuscript proposes a structured and effective method for detecting checkerboard corners in noisy images, especially relevant for high-resolution image stitching and camera calibration tasks. The method includes preprocessing, initial corner extraction, consistency-based filtering, corner grid generation, and completion. The proposed approach is validated through comparative experiments and shows strong performance under various lighting conditions.
- While the method performs well, many components (e.g., edge detection, corner filtering, periodicity analysis) are based on well-known techniques. The novelty lies in how these components are integrated, but this is not made explicit. Add a subsection clearly outlining the differences and improvements compared to references [12], [13], and [15].
- The manuscript lacks any analysis of computational complexity or runtime performance, which is important for deployment in real-time or industrial systems. What is the runtime on high-resolution images (e.g., stitched images)? Can the method support real-time processing? Include a section discussing computational cost, runtime comparisons, or scalability.
- All experiments use the same type of calibration board (17×19 CG-2 checkerboard). This may limit the generalizability of the findings. How does the method perform with different checkerboard sizes or formats, or in the presence of partial occlusion or distortion?
- While visual examples of false corners are presented, there is little quantitative analysis of types and causes of false positives or false negatives. Add an error analysis section to classify and count false detections or analyze edge cases where the algorithm fails.
- Can your method handle cases where part of the checkerboard is occluded or out of frame?
- How sensitive is the algorithm to incorrect input about the number of expected corners?
- How does the method perform under non-uniform illumination or shadows (i.e., local lighting artifacts)?
- Have you considered integrating this method into standard calibration toolkits (e.g., OpenCV) and comparing it directly to their built-in functions?
Comments on the Quality of English Language
Overall, the language is understandable but could be improved with minor polishing for clarity and flow.
Some figure captions are vague. For example: “Figure 4. These are the images related to corner detection” could be revised to “Figure 4. Corner detection pipeline: raw points, centroid calculation, and false positive examples.”
Reviewer 4 Report
Comments and Suggestions for Authors
This paper tries to detect checkboard in noisy stitched for camera calibration. The proposed method is compared with some conventional methods in Ref.12, 13, 15 on one stitiched image under different lighting condition to show its feasibility and advantage s.
However, since the proposed method is for special calibration case, the authors should make this background more clear thus letting the readers to know what the differences between the special situation and general scenes are and what difficulty it is in this situation. Or else the readers cannot understand why the authors study the chessboard detection when there are already mature detection algorithms available.
Other questions:
1. In Eq.8, the threshold for σ may vary for different images, requiring specific adjustments. Then if the threshold is not set properly and there are not enough four neighboring corners found, what will happen?
2. In Fig.5, there is not P_1 point there.
3. In Fig.8, it can be seen that the chessboard is small in the FOV. A stage is used to move the chessboard. Why not use a bigger chessboard?
4. In Fig.6 and Fig.7, there are red dots and green dots. What do they mean? There should be labels explaining them.
5. There are threshold parameters which are related to the concrete images and vary under different images. This may hinder the automatic calibration since many chessboard images will be taken for camera calibration.
6. The detection speed is not introduced.
6. For X point detection, there are more methods can be referenced, like:
Meng, C., Wang, Q., Wu, L., Guan, S.,Wu, Y., Wang, T.: A fast x-corner detection method based on block-search strategy.
Advances in Mechanical Engineering 11(3),1687814019834149 (2019)
Wang, J., Ji, X., Zhang, X., Sun, Z., Wang, T.: Real-time robust individual x point localization for stereoscopic tracking. Pattern
Recognition Letters 112, 138–144 (2018)
Round 2
Reviewer 2 Report
Comments and Suggestions for Authors
My comments were taken into account
The article is of high quality, and I recommend it for publication. However, I have some concerns that it may not find its audience in this journal.
Reviewer 4 Report
Comments and Suggestions for Authors
The authors have made improvement to the paper. The experiment showed the chessboard detection in noisy image is better than compared methods. However, it is not clear yet why a stitched checkboard image is used. If a chessboard is stitched from multiple cameras, I think it cannot be used to calibrate any camera. Or in such case, how to calibrate multiple cameras with a stitched chessboard image is not clear. Therefore, the motivation is still not clear.
- Eq.10 is obviously wrong.
- What is the resolution of the images in Figure 9? What is the relation with the camera's resolution used in the experiment?
- Figure 10 showed the issues of other detection methods, but each image with a different method. They cannot be compared. There should be each image result with all methods.
- English expression should be polished by English-spoken people.
